



# Climatology, sources, and transport characteristics of observed water vapor extrema in the lower stratosphere

Emily N. Tinney [1] and Cameron R. Homeyer [1]

[1]School of Meteorology, University of Oklahoma, Norman, OK, USA

**Correspondence:** Emily Tinney (emily.tinney@ou.edu)

**Abstract.**

Stratospheric water vapor ($H_2O$) is a substantial component of the global radiation budget, and therefore important to variability of the climate system. Efforts to understand the distribution, transport, and sources of stratospheric water vapor have increased in recent years, with many studies utilizing long-term satellite observations. Previous work to examine stratospheric $H_2O$ extrema has typically focused on the stratospheric overworld (pressures $\leq$ 100 hPa) to ensure the observations used are truly stratospheric. However, this leads to the broad exclusion of the lowermost stratosphere, which can extend over depths more than 5 km below the 100 hPa level in the midlatitudes and polar regions and has been shown to be the largest contributing layer to the stratospheric $H_2O$ feedback. Moreover, focusing on the overworld only can lead to a large underestimation of stratospheric $H_2O$ extrema occurrence. Therefore, we expand on previous work by examining 16 years of Microwave Limb Sounder (MLS) observations of water vapor extrema ($\geq$ 8 ppmv) in both the stratospheric overworld and the lowermost stratosphere to create a new lower stratosphere climatology. The resulting frequency of $H_2O$ extrema increases by more than 300% globally compared to extrema frequencies within stratospheric overworld observations only, though the percentage increase varies substantially by region and season. Additional context is provided to this climatology through a backward isentropic trajectory analysis to identify potential sources of the extrema. We show that, in general, tropopause-overshooting convection presents as a likely source of $H_2O$ extrema in much of the world, while meridional isentropic transport of air from the tropical upper troposphere to the extratropical lower stratosphere is also possible.

## 1 Introduction

The troposphere and stratosphere are fundamentally different in their composition of atmospheric trace gases. For example, while abundant in the troposphere, water vapor ($H_2O$) in the stratosphere is uniformly low. In the lower stratosphere (LS), however, the per molecule radiative forcing of $H_2O$ is maximized, where even small increases (on the order of $<$ 1 ppmv) can lead to substantial surface warming (Solomon et al., 2010; Dessler et al., 2013; Wang et al., 2017). An understanding of the sources and controls of stratospheric water vapor is therefore essential to improved understanding of the climate system. Specifically, it is vital to understand processes that facilitate the exchange of air across the tropopause (stratosphere-troposphere exchange; STE), as this can significantly and rapidly alter the composition and therefore radiative forcing of the upper troposphere and lower stratosphere (UTLS).





The LS can be categorized into two separate regions: the extratropical lowermost stratosphere (LMS) and the stratospheric overworld. The stratospheric overworld is conventionally defined where $\theta \geq 380$ K, such that isentropes of the stratospheric overworld remain above the tropopause globally (e.g., Hoskins, 1991; Holton et al., 1995; Stohl et al., 2003). The remaining portion of the stratosphere is the LMS, which lies above the extratropical tropopause but below the 380 K isentrope (i.e. below the height of the tropical tropopause). Therefore, the total LS can be thought of as the combination of the LMS and the lower part of the stratospheric overworld. The concentration of $H_2O$ in the overworld is strongly correlated to and controlled by tropical tropopause temperatures, via the freeze-drying of air across the tropical tropopause as part of the ascending branch of the Brewer–Dobson circulation (e.g., Randel and Park, 2019; Mote et al., 1996). Alternatively, $H_2O$ in the LMS is impacted by both the downwelling branch of the Brewer–Dobson circulation and by frequent STE, specifically troposphere-to-stratosphere transport (TST; Holton et al., 1995; Stohl et al., 2003), though the contributions of specific processes is still not well understood.

At larger scales, enhancements in LMS $H_2O$ concentrations can be linked to isentropic cross-tropopause transport. So-called "tropospheric intrusions" are driven by poleward Rossby wave breaking events and transport tropical upper troposphere air to the extratropical LMS across the tropopause break near the subtropical jet (Pan et al., 2009; Homeyer et al., 2011; Homeyer and Bowman, 2013; Ploeger et al., 2013; Langille et al., 2020). Note that while a small population of these events have been shown to substantially moisten the LMS, tropospheric intrusions are frequently related to decreases in LMS $H_2O$ (Schwartz et al., 2015). Large-scale cross-tropopause transport can also be facilitated by isentropic assent along the warm conveyor belts of midlatitude cyclones, which has been shown to transport $H_2O$ and boundary-layer pollutants into the LMS (Roiger et al., 2011; Stohl, 2001; Wernli and Bourqui, 2002). Isentropic transport related to monsoon dynamics — which is intrinsically linked with smaller-scale monsoon convection — has also been shown to substantially contribute to LMS $H_2O$ enhancements (e.g., Randel et al., 2010; Pan et al., 2016; Honomichl and Pan, 2020; Pan et al., 2022).

Tropopause-overshooting convection typically results in the most extreme stratospheric hydration. Both regional and global climatologies of deep convection show that convection overshoots the extratropical tropopause (and occasionally reaches the stratospheric overworld) relatively frequently over land, especially in the Americas (Solomon et al., 2016; Cooney et al., 2018; Liu and Liu, 2016; Liu et al., 2020; Homeyer and Bowman, 2021). While some studies identify a minimal role of convective contributions to stratospheric water vapor ($\sim$10%), these are typically restricted in focus to tropical convection and impacts on the stratospheric overworld (e.g., Dauhut and Hohenegger, 2022; Ueyama et al., 2023, and references therein). Studies that focus on convection within extratropical environments, subtropical environments, and monsoon regions often show substantial contributions from convection to the LMS $H_2O$ concentration locally (Hanisco et al., 2007; Dessler and Sherwood, 2004; Smith et al., 2017; Jensen et al., 2020; Tinney and Homeyer, 2021; Gordon and Homeyer, 2022; Phoenix and Homeyer, 2021; Homeyer et al., 2014; Hegglin et al., 2004; Mullendore et al., 2005; Schwartz et al., 2013; Werner et al., 2020; O'Neill et al., 2021). Overall, the contributions of any specific process to the stratospheric $H_2O$ budget, especially deep convection, remains a topic of scientific debate.

An important instrument that has been frequently employed in studying global LS $H_2O$ is NASA's Microwave Limb Sounder (MLS). For example, Schwartz et al. (2013) and Werner et al. (2020) use MLS observations to assess the global distribution of high $H_2O$ concentrations at a pressure level of 100 hPa, which is commonly found at a similar level to the 380 K isentrope.





Both studies show that high $H_2O$ concentrations ($\geq$8 ppmv) are most frequent in monsoon-related active-convection regions and therefore contribute to the growing body of evidence suggesting that convection is a substantial contributor to LS $H_2O$, especially at a regional level. However, such studies do not evaluate the frequency of $H_2O$ enhancements in the LMS, which can encompass a layer 5 km or deeper below the 100 hPa and 380 K levels (Holton et al., 1995). Higher MLS pressure levels have not been considered in previous studies due to large latitudinal and seasonal variations in tropopause heights complicating the diagnosis of LMS layers. Unfortunately, this choice is likely to lead to substantial underestimations of both the frequency and magnitude of enhanced LS $H_2O$ concentrations. The potential for underestimation of convection-driven extrema specifically is expected to be impacted greatest since convection-driven enhancements are typically confined to only a few km above the tropopause (e.g., Tinney and Homeyer, 2021).

Therefore, this study intends to expand upon previous work by examining 16 years (2005–2020) of MLS $H_2O$ observations to create a climatology of $H_2O$ extrema in both the lowermost and overworld stratosphere. To achieve this, we use reanalysis data to diagnose whether individual layers in an MLS profile are stratospheric, allowing for accounting of observed LMS $H_2O$ extrema for the first time. Additional context is provided to these observations through an isentropic back-trajectory analysis of common transport pathways and discussion of the potential roles of large-scale vs. convective sources.

## 2 Data and methods

### 2.1 Reanalysis

Three-hourly assimilations of the global atmosphere are employed from the NASA Modern-Era Retrospective Analysis for Research and Applications, version 2 (MERRA-2; Gelaro et al., 2017). Temperature, pressure, potential vorticity (PV), and wind fields are used in this study. MERRA-2 lapse-rate tropopause (LRT) heights and pressures are calculated according to the World Meteorological Organization (WMO) definition (World Meteorological Organization, 1957). MERRA-2 is available from 1979–present on an approximate 0.5° x 0.625° longitude-latitude grid with 72 vertical model levels, which corresponds to ∼1.1 km vertical resolution in the UTLS.

### 2.2 Global $H_2O$ observations

Measurements of $H_2O$ in the UTLS are sourced from the Earth Observing System (EOS) Microwave Limb Sounder (MLS) v5.0x dataset. The MLS is aboard the Aura spacecraft as part of the NASA A-train constellation of sun-synchronous satellites, and has equator crossing times of 0130 and 1330 LT. The instrument performs a continuous vertical scan of the atmosphere (surface–90 km) in the forward direction of orbital motion, completing ∼3600 profiles per day with a 1.5° along-track separation between each scan (Livesey et al., 2020). Concentrations of 16 different trace gases have been collected globally by MLS since August 2004. The MLS retrieval range of $H_2O$ is 316–0.001 hPa, with measurements at 12 levels per decade of pressure in the UTLS. The precision, accuracy, horizontal resolution and vertical resolution of the measurement varies with height, ranging from 4–65%, 4–25%, 168–400 km, and 1.3–3.5 km, respectively, for pressures 316–1.0 hPa before degrading



at lower pressures. Only MLS layers with pressures of 147 hPa and below are analyzed here, where the precision and accuracy of the measurement are more suitable for this study. The data is quality controlled following the recommendations of Livesey et al. (2020). The MLS v5.0x has a number of improvements from previous data versions, including partial amelioration of a

calibration-related drift in the $H_2O$ measurement.

## 2.3 Stratospheric $H_2O$ extrema identification

MLS observations from 2005–2020 are utilized in conjunction with MERRA-2 data to asses the frequency of $H_2O$ extrema in the stratosphere. MERRA-2 LRT pressure, PV, and potential temperature ($\theta$), are linearly interpolated in space and time to each MLS profile location, and logarithmically interpolated vertically to the individual layers of each MLS profile. These

atmospheric parameters are then employed to diagnose whether any individual MLS layer is located in the stratosphere. Due to the relative thickness of MLS layers and potential uncertainties in LRT identification, additional evaluation is required to identify stratospheric MLS layers free of tropospheric contamination (which would result in artificially high frequencies of stratospheric $H_2O$ extrema, especially in the LMS). Therefore, based on rigorous testing and evaluation, we require that layers meet a set of three criteria to be classified as wholly stratospheric: (1) $PV \geq 6$ PVU, (2) $log(P_{MLS}) \leq log(P_{LRT}) - 0.075$

(i.e. the layer must be at least $\sim 1$ km above the LRT), and (3) $\theta \geq 340$ K. While these criteria are applicable in the mid- and high-latitudes, they are inappropriate for application to tropical profiles due to PV converging to zero in this region. Therefore, we also consider MLS layers to be stratospheric if $\theta \geq 380$ K. As an upper limit for layers to be included in the analysis, we additionally require that layers have a $\theta \leq 450$ K to restrict the analysis to lower stratosphere layers only. This set of stringent criteria allows us to analyze stratosphere-only observations and ensure that tropospheric contamination is minimized.

For analysis, all identified stratospheric MLS layers are collected in $5°$ latitude-longitude bins. The frequency of $H_2O$ extrema (exceeding a given threshold) in each bin are then calculated. To quantify how inclusion of the LMS impacts the distribution and frequency of extrema identification, the same binning process is completed for stratospheric overworld ($\theta \geq 380$ K) observations only. Due to seasonal variation in the frequency, location, and magnitude of stratospheric $H_2O$ extrema, analysis is conducted separately for DJF (December, January, February), MAM (March, April, May), JJA (June, July, August),

and SON (September, October, November) when necessary.

## 2.4 Trajectory analysis

To provide context to the LMS $H_2O$ extrema climatology, large-scale transport characteristics are explored via isentropic trajectory analyses. Trajectories are initialized at the latitude, longitude, and $\theta$ of stratospheric $H_2O$ extrema that occur within eight identified high-frequency regions shown in Fig. 1. Using the TRAJ3D trajectory model (Bowman, 1993; Bowman and

Carrie, 2002), particles are advected backward in time using MERRA-2 winds for up to 10 days, with positions saved every 6 hours along the trajectory path. Two-dimensional (latitude-longitude) frequency distributions of trajectory particle locations at multiple time intervals are used to identify common pathways to regions of frequent extrema. Given the MERRA-2 spatiotemporal resolution and wind field uncertainties, horizontal displacement errors of individual trajectories are expected to be $\sim 60$ km per day (Bowman et al., 2013; Stohl et al., 1995), but these errors are largely irrelevant for examining the bulk transport





behavior sought here. Evaluating the recent history of identified $H_2O$ extrema air masses helps to provide context for their potential (or likely) sources.

## 2.5   Observations of convection

Observations of tropopause-overshooting convection are sourced from NASA's Global Precipitation Measurement (GPM) mission. The GPM core satellite was launched in 2014 and is able to measure precipitation characteristics in three dimen-

sions, allowing for the detection of precipitation features from the tropics to the middle- and high-latitudes (Hou et al., 2014; Skofronick-Jackson et al., 2017; Nesbitt et al., 2000; Liu et al., 2008). These precipitation features can be used in combination with tropopause altitudes to identify overshooting convection. We use an extended record (2015–2020) of GPM overshoots that was originally produced for and analyzed in Liu et al. (2020), which has been updated to use the newer ECMWF Reanalysis Version 5 (ERA 5) LRT as a reference (Hersbach et al., 2020). Any precipitation feature (radar echo >20 dBZ) found at an

altitude above the ERA5 LRT altitude is classified as an overshoot. We use the resulting seasonal geographic distributions of overshoot frequency to provide context for the transport analysis in this study.

## 3   Results

The analysis presented here was completed for three different thresholds of $H_2O$ extrema (8, 10, and 12 ppmv). The frequency of extrema identification decreases as the threshold increases (as expected), but the choice of threshold does not play a notice-

able role in the global distribution and relative frequencies of the extrema. Only the results for $H_2O$ extrema exceeding 8 ppmv are presented here, as this is the most commonly used extrema threshold in prior work.

### 3.1   Extrema frequency

The frequency of $H_2O$ extrema in the total LS (overworld + LMS) and the overworld only are shown in Fig. 1a,b. Over most of the world, $H_2O$ concentrations exceeding 8 ppmv in the stratospheric overworld occur less than 0.25% of the time. There

are six notable geographic features where the frequency of extrema maximizes which we highlight and subjectively classify into regions here: Central and Eastern Asia (CEA), the North Pacific (NP), the South Pacific (SP), the Gulf of California (GC), North America and the North Atlantic (NA), and finally South America and the South Atlantic (SA). The maximum frequency of overworld $H_2O$ extrema in each of these regions varies from ∼0.25–1.25%. The CEA feature is the most pronounced in its spatial extent and magnitude, followed closely by the NA feature.

When this analysis is extended to include the LMS, the magnitude and spatial extent of nearly every feature increases, although the strength of the frequency change is variable across the domain (Fig. 1a). The NA, GC, NP, and SP maxima experience the greatest increases in frequency magnitude, exceeding 2% in some locations which is more than double that of their overworld counterparts. The SA feature displays a modest increase in frequency, with a maximum frequency of ≤1.25%. Notably, the magnitude of extrema in the CEA region is minimally impacted by the inclusion of the LMS. This results in the

central Asia maximum being one of the least pronounced features in the total LS, despite being the dominant region in the

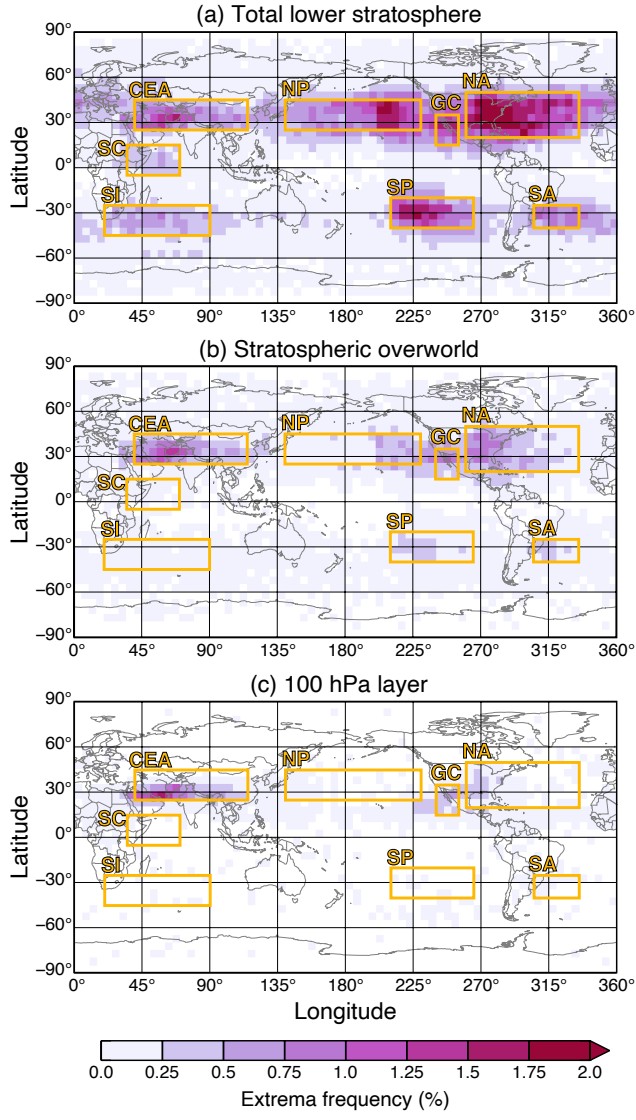

**Figure 1.** Binned frequency of H$_2$O extrema ($\geq$ 8 ppmv) as observed by MLS for (a) all layers classified as stratospheric, (b) overworld layers only, and (c) the 100 hPa layer only. Eight local maxima are classified into regions (gold) for further analysis.

overworld only analysis. There are also two additional maxima that become apparent with the inclusion of the LMS: along and just east of the Somalian Coast (SC), and over the Southern Indian Ocean (SI). These features were not detectable in the overworld only analysis where, like in much of the rest of the world, the occurrence of H$_2$O extrema did not exceed a frequency of 0.25%. However, these maxima become comparable to the CEA feature in the total LS analysis, with extrema frequencies reaching up to ~1%. In addition to changes in the magnitude of H$_2$O extrema frequencies in the total LS analysis for most features, the spatial extent of most features increases as well. Specifically, the features tend to be elongated zonally from their





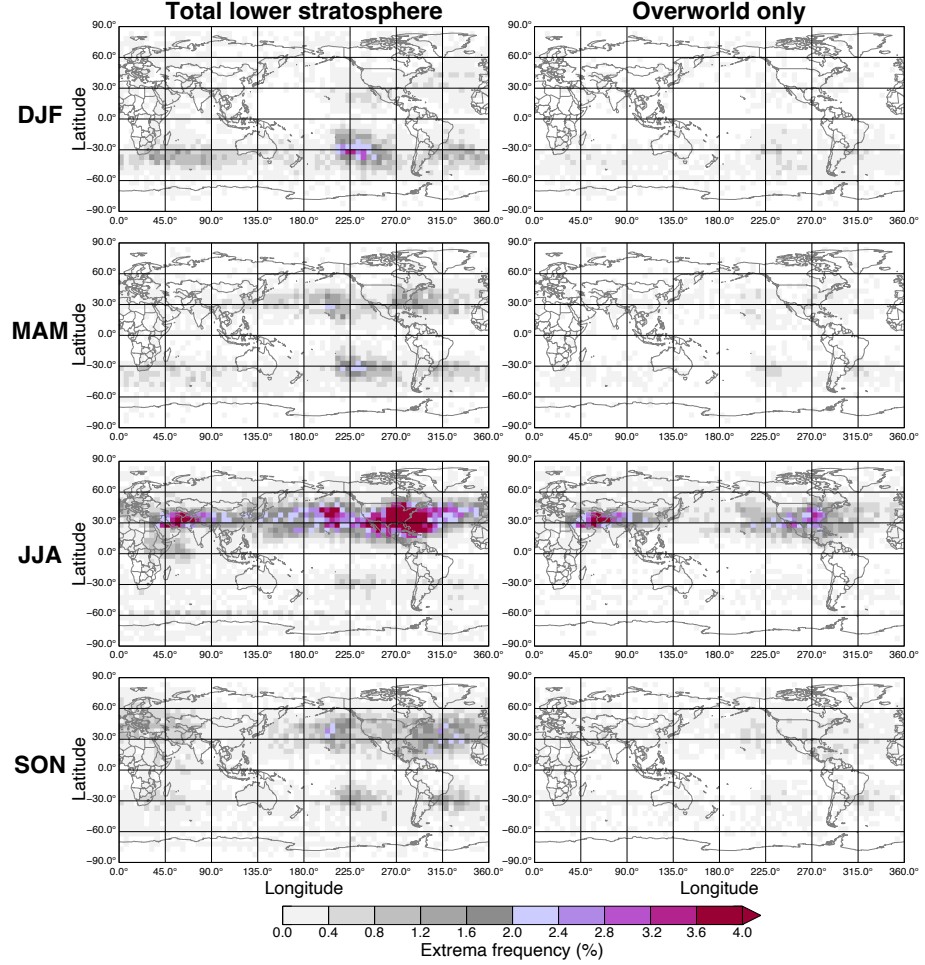

**Figure 2.** Binned frequencies of MLS $H_2O$ extrema ($\geq 8$ ppmv) separated seasonally into December, January, and February (DJF; top row), March, April, and May (MAM; second row), June, July, and August (JJA; third row), and September, October, and November (SON; bottom row) for (left) all stratospheric layers and (right) stratospheric overworld layers only.

position in the overworld. This can most clearly be seen in the NA and SA features extending eastward over the Atlantic, and the NP feature extending westward to far-eastern Asia. Note that the regions were subjectively chosen based on the locations of maxima in the total lower stratosphere analysis.

The prominent features over North America (both the NA and GC regions), Asia (the CEA region), and South America (the SA region) have been seen in previous studies of MLS $H_2O$ extrema at the 100 and 82.5 hPa levels, and have been linked to convective sources associated with the monsoon anticyclone circulations on these continents (e.g., Werner et al., 2020; Schwartz et al., 2013). Alternatively, the maxima over the north and south Pacific (the NP and SP regions), and over the Indian Ocean (the SC and SI regions) have never been identified. This, combined with no obvious local convective sources of said



features, may lead to some concerns that this result could be a nonphysical artifact of or error in the analysis. For this reason, we also apply our analysis to the 100 hPa layer only to allow for a comparison to previous work (Fig. 1c). These results are nearly identical to those shown in Werner et al. (2020) and Schwartz et al. (2013), with minor differences likely accounted for by the length of the MLS record used, bin sizes, and previous choices to exclude certain anomalous events that were not made here. Most importantly, the NP, SP, SC, and SI are not found in our 100 hPa only analysis. The similarity of the analysis presented here to the results in Schwartz et al. (2013) and Werner et al. (2020) provide confidence that the previously unseen features are not due to analysis error, but rather due to the inclusion of additional MLS layers that can be classified as stratospheric. However, the presence of the LMS in the deep tropics where the SC region is located is — by definition — non-existent, which leads to a question of how the total LS analysis indicates a local maxima over this region when it is not present in the stratospheric overworld. This is investigated further in the transport analysis below.

The seasonal breakdown of the $H_2O$ extrema patterns are shown in Fig. 2. In the Northern Hemisphere, JJA dominates the annual cycle in both the total LS and the overworld. In the NA and NP regions, the frequency of $H_2O$ extrema in JJA far surpasses that of any other season, with more than 4% of total LS observations exceeding 8 ppmv. The westward extent of NP maxima seen in Fig. 1 is even more evident when restricted to JJA only. MAM and SON have modest contributions to Northern Hemisphere extrema and are most substantial over the NP and NA regions, while DJF (boreal winter) frequencies are < 0.4% across nearly the entire Northern Hemisphere. The significance of the Asian Monsoon Anticylone is made apparent in the total LS seasonal analysis where the CEA maxima is pronounced in JJA, while other features — such as NA maxima — are present in all seasons except for DJF. In the stratospheric overworld, however, locations over CEA and the NA regions exceed an extrema frequency of 0.8% in JJA only.

In the Southern Hemisphere, DJF (austral summer) has the most prominent contribution to both LS and overworld extrema, though the overall annual cycle is far less clear than that of the Northern Hemisphere. The SP region is the dominant feature of the DJF analysis, with frequencies exceeding 2–3% throughout the region. The SI and SA maxima are also noticeable in the DJF total LS analysis, though their frequencies remain below 1.6%. Similar to their contributions in the Northern Hemisphere, MAM and SON feature modest frequencies of $H_2O$ concentrations exceeding 8 ppmv in the Southern Hemisphere, with the maximum over South America being the only notable feature in addition to that over the Pacific. Finally, Southern Hemisphere extrema in JJA (austral winter) are exceedingly rare.

## 3.2 Transport characteristics

To provide context to the extrema observations described above, we investigate the recent transport behavior of all LS $H_2O$ extrema located in the eight regions identified in Fig. 1 via an isentropic backwards trajectory analysis for the season in which the feature is most pronounced. This analysis serves as a compliment to the extrema climatology presented above and allows us to investigate potential sources of extreme LS $H_2O$. We show here and discuss in detail the statistical transport for a well-known $H_2O$ frequency maximum (the CEA region), and two unexpected maxima (the NP and SC regions). The transport analyses for the remaining regions are located in the appendix.



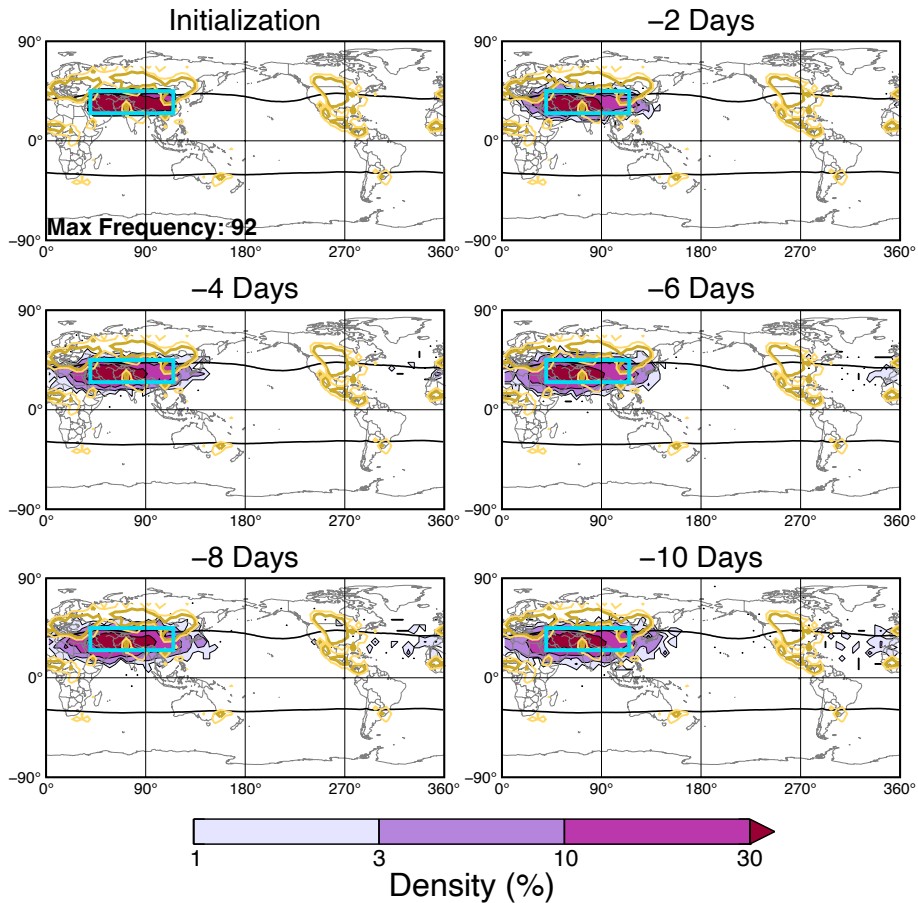

**Figure 3.** Isentropic backwards trajectory analysis for $H_2O$ extrema in the CEA region (blue box) in JJA. The normalized density of trajectories at initialization and at 2, 4, 6, 8, and 10 days prior is shown by the color-fill, with the maximum density value given in the initialization panel. The seasonal frequency of tropopause-overshooting convection as detected by GPM is given by the golden contours at intervals of $5 * 10^{-5}$ overshoots per observation (lighter gold) and $10 * 10^{-5}$ overshoots per observation (darker gold). The seasonal average tropopause break for each hemisphere is indicated by the solid black line.

The statistical transport behavior of $H_2O$ extrema located in the CEA region during JJA is shown in Fig. 3. Throughout the 10 day history, the vast majority of trajectory particles remain over Asia indicating that the extrema air has been confined within the summertime Asian Monsoon anticyclone throughout its recent history. As expected, and consistent with previous studies (e.g., Bergman et al., 2013; Khaykin et al., 2022), this demonstrates that the frequent high LS $H_2O$ concentrations over this region are related to a combination of monsoon dynamics and convection. It is important to note, however, specific convective moistening of the particles along the trajectory path may have occurred before or at any time during the preceding 10 day period, as convective transport is not captured by these large-scale isentropic trajectories. For the NP maxima, transport is largely zonal along the subtropical jet axis (Fig. 4). At 4 days prior to the extrema observation, the highest density area of



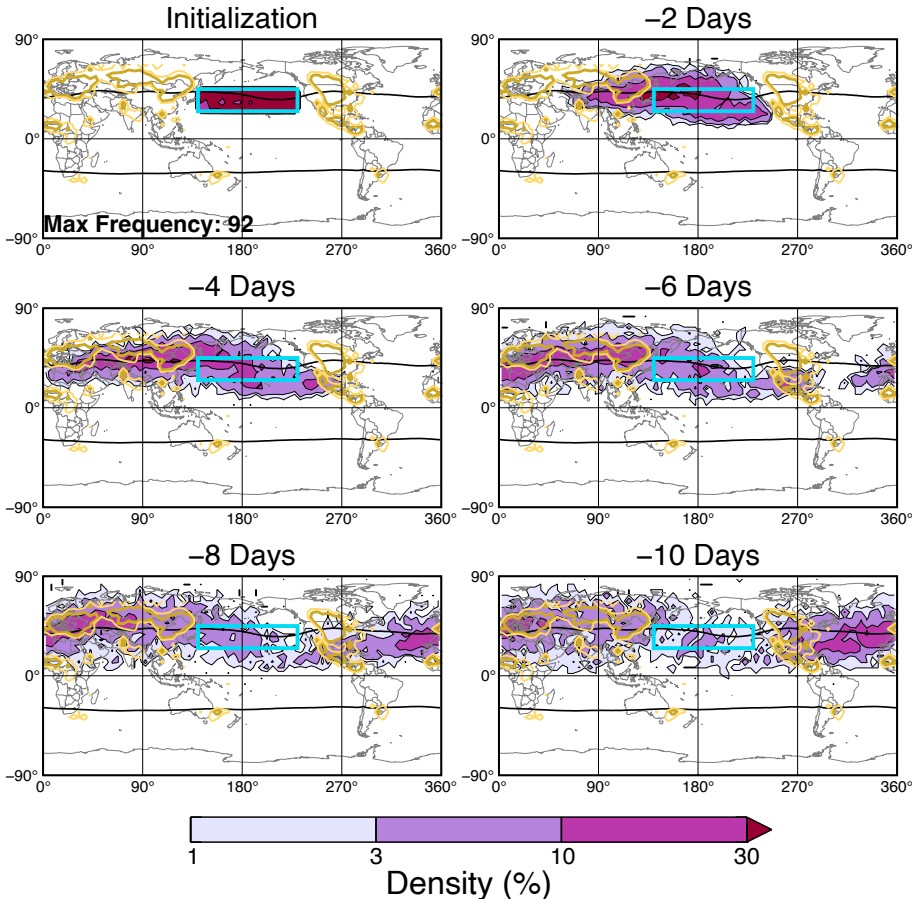

**Figure 4.** As in Fig. 3, but for the NP region in JJA.

trajectory particles is located over active overshooting convection areas across Asia, Siberia, and southern Russia as observed by GPM, suggesting that convective moistening is a likely contributor to these extrema. A smaller, but still substantial, portion of trajectories trace back to central American convection 4 to 6 days prior. As demonstrated in Figs. 1 and 2, the frequency of $H_2O$ extrema in the eastern half of the defined NP region is approximately double that of the western half. This transport analysis

suggests that central American convection related to the North American Monsoon Anticylone is at least partly responsible for the high frequency of extrema over the eastern North Pacific. Another potential source for high LS $H_2O$ concentrations over the NP worth investigating would be poleward Rossby wave breaking transport of tropical/subtropical upper troposphere air. The north Pacific is a location of frequent Rossby wave breaking (Homeyer and Bowman, 2013), and poleward wave breaking has the potential to transport relatively moist, tropical upper tropospheric air into the lowermost stratosphere and contribute to

this maximum (Langille et al., 2020). However, the lack of substantial meridional transport from the tropics (i.e., equatorward of the average tropopause break latitude) related to the observed extrema, outside of the aforementioned path from central





America, suggests that this method of stratospheric hydration may be limited when it comes to $H_2O$ concentrations exceeding 8 ppmv.

Finally, the transport history of the SC local maximum is shown in Fig. 5. As mentioned above, the existence of relatively high frequencies of $H_2O$ extrema in the SC region in the total LS analysis, but not the overworld only analysis, is theoretically impossible, as the LMS does not exist in the deep tropics. The transport behavior indicates that this air largely originates from southeast Asia as recently as two days prior, and was located within the monsoon circulation for the preceding 10 days. The path of these trajectories largely resembles equatorward wave breaking of midlatitude LMS air along the eastern portion of the monsoon anticyclone shown in previous studies (e.g., Konopka et al., 2010). The transport of this air into the deep tropics would retain some characteristics of its source region for up to 1 week, namely higher PV and potential temperature, which is likely what allows for this air to meet the threshold requirements set here and identify it as LMS though it is encompassed by tropical upper troposphere air. Additionally, LRT altitudes in this region are frequently identified lower than in other regions located along the same latitude band (not shown), again suggesting a modification confined to this region due to monsoon dynamics.

To provide additional insight into the potential sources of LS $H_2O$ extrema, we can analyze the cross-tropopause transport nature of the isentropic trajectories to assess the likelihood of large-scale moistening (rather than delivery by tropopause-overshooting convection). As a proxy for large-scale isentropic TST, the percentage of trajectories that spent at least 72 of the 120 hours prior to extrema observation within the troposphere in each season is shown in Fig. 6. The seasonal variation of large-scale TST at any given location appears minimal. However, it is important to note that for each season, data is only shown for bins with at least 20 initialized trajectories, which could obscure seasonal variation from this analysis. In general, locations over the south Pacific, southern Indian Ocean, the Somalian coast, and the Asian Monsoon region more frequently ($\geq$60%) indicate recent large scale TST, while the northern Pacific and North America have much lower large-scale transport percentages (<40%).

The higher frequency of large-scale TST over the Asian monsoon region (>80% in some places) is consistent with recent studies that have shown the importance of monsoon dynamics in stratospheric moistening over Asia, where monsoon convection often moistens the upper troposphere but additonal monsoon-driven isentropic cross-tropopause transport is required to extend these impacts to the stratosphere (e.g., Randel et al., 2010; Pan et al., 2016; Honomichl and Pan, 2020; Pan et al., 2022). Alternatively, the lower percentages common throughout the rest of the Northern Hemisphere subtropics and extratropics suggest that direct convective moistening via overshooting is the primary driver of these extreme concentrations. In some locations, like over North America extending eastward into the North Atlantic, this adds to the body of work which have shown that convection over North America is particularly capable of moistening the lowermost stratosphere (e.g., Randel et al., 2012; Tinney and Homeyer, 2021).

On the other hand, the low frequencies of large-scale TST for the summertime band of extrema from 180–225°E Longitude over the northern Pacific is somewhat surprising given that this is a location of frequent Rossby wave breaking in boreal Summer (Homeyer and Bowman, 2013). However, this is in line with the analysis shown in Fig. 4 which has a lack of meridional transport from the tropics outside of a pathway from summertime transport from central American convection, which suggests





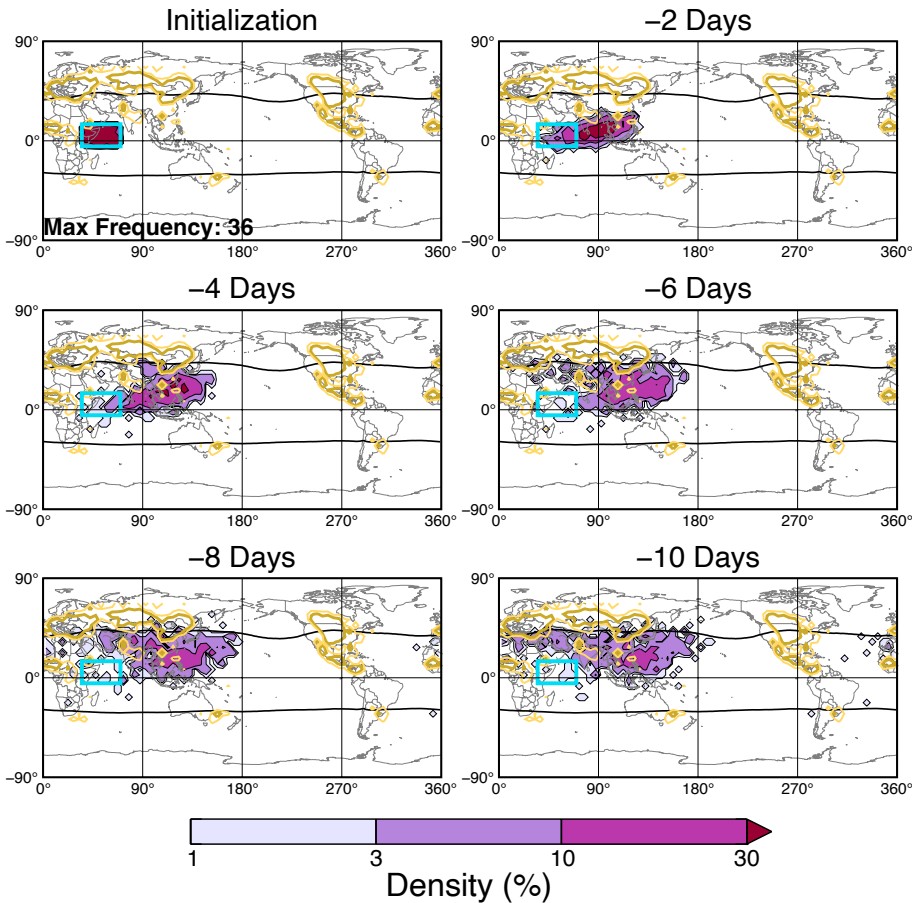

**Figure 5.** As in Fig. 3, but for the SC region in JJA.

poleward Rossby wave breaking is not a substantial contributor for LS $H_2O$ concentrations greater than 8 ppmv. The significance of the contribution of Rossby Wave breaking events to stratospheric $H_2O$ concentrations has been debated in previous work (e.g., Ploeger et al., 2013). The analysis above suggests that while horizontal transport events between the tropical upper troposphere and extratropical LS via Rossby wave breaking may be common in this location, the air involved in associated TST is not moist enough to substantially contribute to the populations of $H_2O$ extrema analyzed here.

### 3.3 Annual cycles in monsoon-related regions

Analysis of the annual cycle of LS $H_2O$ extrema in monsoon-related regions provides additional insight to the impact of monsoon circulations and dynamics on $H_2O$ extrema. In particular, we focus in the Asian Monsoon Anticyclone (AMA; 20°N – 40°N, 30°E – 130°E), the North American Monsoon Anticyclone (NAMA; 20°N – 45°N, 230°E – 290°E), and the South American Monsoon Anticyclone (SAMA; 20°S – 40°S, 260°E – 320°E). Note that these region boundaries are different



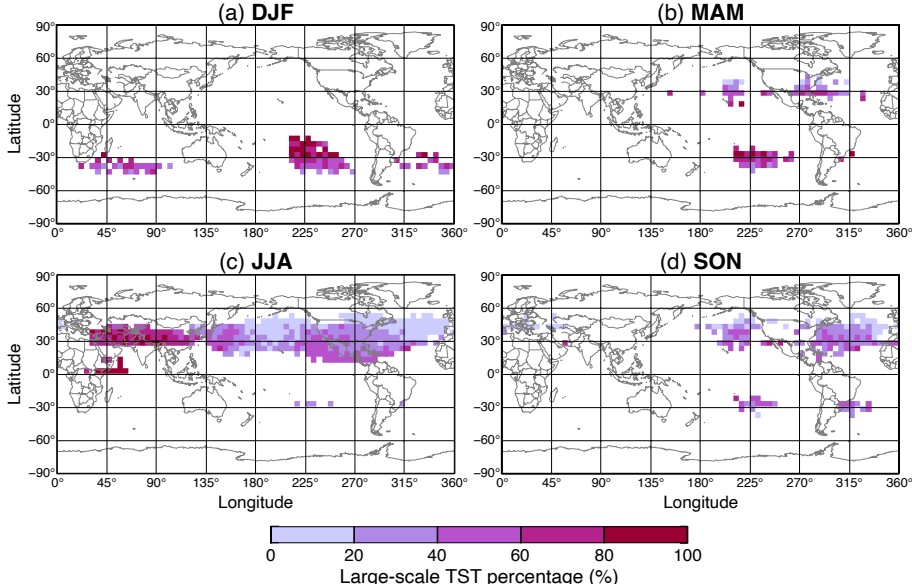

**Figure 6.** The binned percentage of $H_2O$ extrema trajectories classified as large-scale TST at their initialized location for (a) DJF, (b) MAM, (c) JJA, and (d) SON. Percentages are only shown for bins with $\geq 20$ observations for the corresponding season.

from those defined and discussed previously, as those were subjectively chosen based on local maxima of LS $H_2O$ extrema frequency and do not necessarily align with the tropopause-level monsoon circulations. These monsoon regions, shown in Fig. 7d, were specifically chosen to encapsulate their associated upper tropospheric anticyclonic circulations as indicated by the 270 climatological mean of hemispheric summer 100 hPa winds in reanalysis (not shown).

While the frequency of $H_2O$ extrema peaks in summer and decreases in winter for each monsoon anticyclone, the characteristics of each cycle vary substantially. Both when normalizing for region size (Fig. 7a) and not (Fig. 7b), the frequency of LS $H_2O$ extrema in AMA and NAMA are an order of magnitude larger than in SAMA at their respective peaks. The SAMA annual cycle is characterized by a broad, shallow peak from October to January (hemispheric Spring and Summer) with a maximum 275 average of $\sim$0.2 observations per gridpoint. For NAMA the occurrence of LS $H_2O$ extrema largely exists between April and October (hemispheric late Spring to early Fall), peaking in August at a maximum average of $\sim$1.7 observations per gridpoint. Alternatively, AMA extrema primarily exist within boreal summer (JJA) and peak in July at $\sim$1.0 observations per gridpoint. From a per gridpoint standpoint, the NAMA region clearly dominates contributions to LS $H_2O$ extrema both in magnitude of the frequency and the longevity, likely in part due to contributions from spring and summertime deep convection over the 280 United States that is unrelated to the monsoon circulation. When comparing the monsoons as a whole and allowing for their size to modulate their contributions, the NAMA region still exhibits the greatest $H_2O$ extrema frequency, though AMA is more comparable in the total number of extrema observations (Fig. 7b). Perhaps even more notable is the disparity between the proportion of total LS versus overworld only extrema in each region. For NAMA and SAMA, the overall overworld contributions to the total LS extrema frequency is less than 50%, while more than 90% of AMA LS $H_2O$ extrema are from the stratospheric



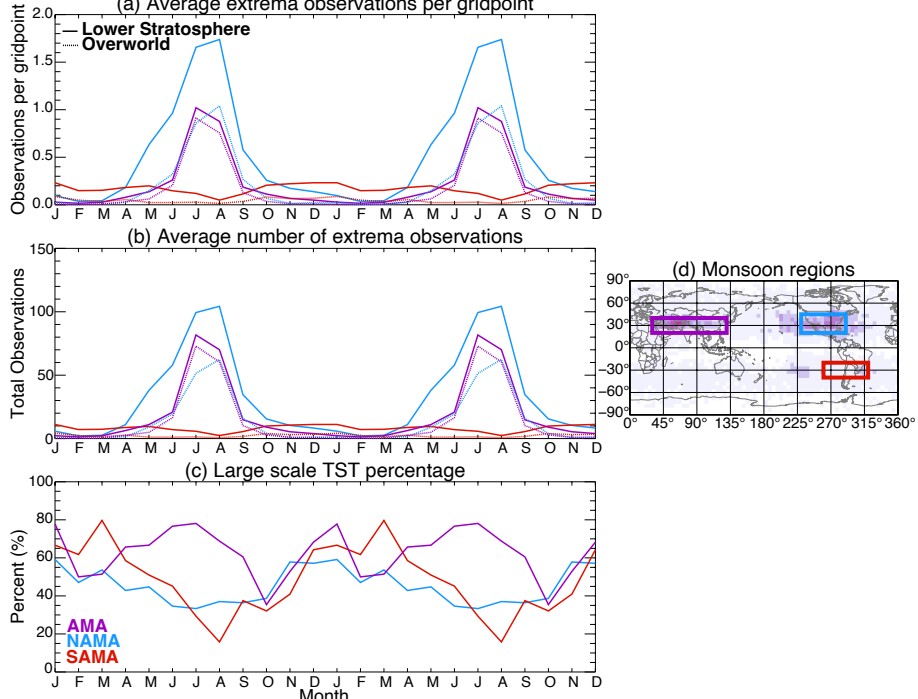

**Figure 7.** Two average annual cycles of (a) the number of $H_2O$ extrema per gridpoint, (b) the average regional number $H_2O$ extrema, and (c) the percentage of extrema with a large-scale transport history are given for the Asian Monsoon Anticyclone (AMA; 20°N – 40°N, 30°E – 130°E; purple), the North American Monsoon Anticyclone (NAMA; 20°N – 45°N, 230°E – 290°E; blue) and the South American Monsoon Anticyclone (SAMA; 20°S – 40°S, 260°E – 320°E; red). For (a) and (b), the number of observations for the total lower stratosphere is given by the solid line and the number of observations in the stratospheric overworld only is given by the dotted line. The region boundaries for each monsoon described above are given in (d).

overworld. This result especially highlights the importance of considering the LMS when assessing the contributions of each monsoon to extreme $H_2O$ concentrations.

Figure 7 also shows the percentage of extrema in each monsoon region whose back trajectory analysis indicates a recent tropospheric origin (as described above). Again, there are substantial differences between the three monsoons. For AMA and SAMA, the large-scale TST percentage peaks during the monsoon season when the frequency of extrema peaks. Alternatively, the NAMA region experiences a minimum in large-scale TST percentage in August when the $H_2O$ extrema frequency peaks, providing more evidence that convection in the NAMA region is uniquely capable of transporting $H_2O$ to the LS without necessitating some additional, larger-scale transport, as has been demonstrated in previous studies (e.g., Randel et al., 2012; Tinney and Homeyer, 2021).



## 4    Conclusions

MLS observations from 2005–2020 were used in conjunction with MERRA-2 reanalysis data to create a climatology of $H_2O$ extrema ($>8$ ppmv) in the stratospheric overworld and in the total LS (overworld + LMS). We show that the frequency and distribution of $H_2O$ extrema in the total LS (0.27% globally) is dramatically different from that of the stratospheric overworld (0.08% globally), revealing that the frequency of LS extrema increases by more than 300% when the LMS is included in analysis. On both a yearly and seasonal basis, the frequency of extrema in the total LS analysis is substantially greater than that 300 of the stratospheric overworld, but the magnitude of the difference varies by region (Figs. 1 and 2).

To provide additional context to this climatology, a statistical transport analysis was conducted by initializing isentropic trajectories at the latitude, longitude, and $\theta$ of $H_2O$ extrema (Figs. 3–5, A1–A5). The transport analysis reveals two main transport patterns: (1) air being traced to or confined within monsoon circulations (i.e. the CEA, SC, NA, and GC regions; Figs. 3, 5, A1, and A3) and (2) largely zonal transport along the tropopause break via subtropical jet streams (the NP, NA, 305 SA, SP, and SI regions; Figs. 4, A1, A2, A4, and A5). For all regions, the large-scale transport pathways indicate that the extrema air can be traced to regions of relatively frequent tropopause-overshooting convection. This analysis also reveals that meridional transport from the tropics to the observed $H_2O$ extrema is infrequent. To further investigate the potential origins of $H_2O$ extrema, the cross-tropopause nature of the isentropic trajectories was also investigated (Fig. 6). The percentage of trajectories classified as being related to large-scale TST is regionally dependent, notably showing low occurrences of large- 310 scale TST over the NP, NA, and GC regions — providing further evidence of convection serving as the major source of $H_2O$ extrema in those regions. In regions where large-scale TST is more frequent, it remains unknown whether convection upstream is coupled to such extrema. Namely, moist air that is transported isentropically to the LS may be related to upstream convective sources that acted to hydrate the upper troposphere prior to the large-scale TST.

Finally, the annual cycles of extrema frequency were investigated for regions encompasing the AMA, the NAMA, and the 315 SAMA (Fig. 7). The LS frequency of $H_2O$ extrema in the AMA and NAMA regions were shown to be an order of magnitude larger than that of the SAMA. Additionally, while the AMA and NAMA have similar overworld extrema frequencies throughout the annual cycle, the magnitude and duration of peak extrema frequencies for the NAMA increases substantially with the inclusion of the LMS in the total LS analysis, compared to a small increase for the AMA. The results presented above highlight the importance of the including the LMS in analyses of LS composition. The frequency, geographic extent, and longevity of 320 extrema are all substantially larger in the total LS analysis compared to the overworld. Additionally, the transport analysis strongly suggests that convection is a substantial contributor to the occurrence of LS $H_2O$ extrema, which may not have been clear if conducted for the overworld only.

The major limitation and challenge for this work arose from the major goal of this study — to expand on previous work through the inclusion of the LMS in analysis of LS composition. Restricting the analysis to stratospheric MLS layers only 325 proved to be a difficult task due to the relatively coarse vertical resolutions of MLS and MERRA-2. To limit contamination from MLS layers whose depth may extend across the tropopause, a series of stringent criteria were put in place and only MLS layers at pressures of 147 hPa were included in this analysis. Despite these efforts, it is possible that upper-tropospheric



H$_2$O could influence parts of the analysis and partially inflate LS extrema frequencies. Alternatively, the stringent criteria may also obscure and prevent truly lower stratospheric layers from being included within this analysis — therefore potentially

undercounting extrema. We emphasize here that the inclusion of the LMS in analyses like that presented here is challenging — but worthwhile — and is important to do in future work that aims to increase understanding of the concentrations and sources of H$_2$O and other trace gases in the LS, especially given the implications for understanding the role of tropopause-overshooting convection in the STE budget.

*Data availability.* All data used in this study are publicly available. MLS and MERRA-2 data were obtained from the NASA GES DISC

(Lambert et al., 2020; Global Modeling and Assimilation Office (GMAO), 2015). The processed GPM precipitation feature data used in this study are available online at http://atmos.tamucc.edu/trmm/data/.

**Appendix A**

The statistical backward trajectory transport analysis described and shown for the CEA, NP, and SC regions in the main text is presented and breifly discussed here for the remaining regions. Figures A1–A3 show back trajectory density maps for H$_2$O

extrema in the NA, SA, and GC regions. These regions are all characterized by rapid tranpsort of extrema observations to active overshooting convection regions upstream and spatially adjacent to the extrema locations, implying that MLS is capturing H$_2$O enhancements from convection at times shortly after the storms. For the NA region, overshooting over the U.S. Great Plains, Gulf of Mexico, and — at longer transport times — the Mediterranean, are likely contributors (with significance in that order). For the SA region, overshooting in Argentina is most likely responsible. For the GC region, overshooting over the Sierra

Madre Occidental in Mexico and the Gulf of Mexico are likely contributors. In contrast with these apparently dominant local convective sources, transport pathways of H$_2$O extrema in the SP region (Fig. A4) are not linked to a clear overshooting source region, but are densely sourced from the equatorward side of the mean tropopause break location. This behavior suggests that much of the H$_2$O extrema in that region are facilitated in part by large-scale TST. It is noted, however, that SP extrema transport bypasses the South Pacific Convergence Zone (SPCZ), which is one of the more globally active convective regions

in DJF (when SP extrema are most common; Vincent, 1994, and references therein). Thus, it is possible that H$_2$O extrema in this region are the result of large-scale transport of UT air hydrated by convection over the SPCZ to the LS over the east Pacific. Finally, transport histories for H$_2$O extrema within the SI region (Fig. A5) indicate rapid linkages to two upstream overshooting convection sources along the mean tropopause break location (i.e., the subtropical jet) within 2–4 days: southern Africa and Argentina.





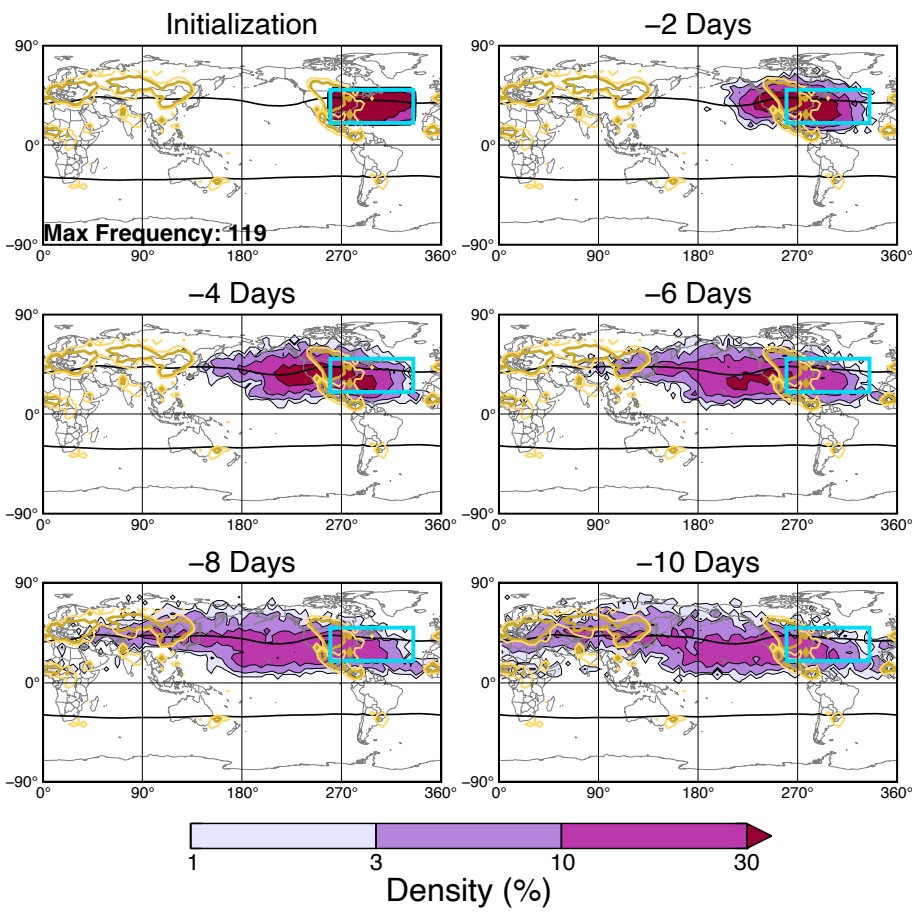

**Figure A1.** As in Fig. 3, but for the NA region in JJA.



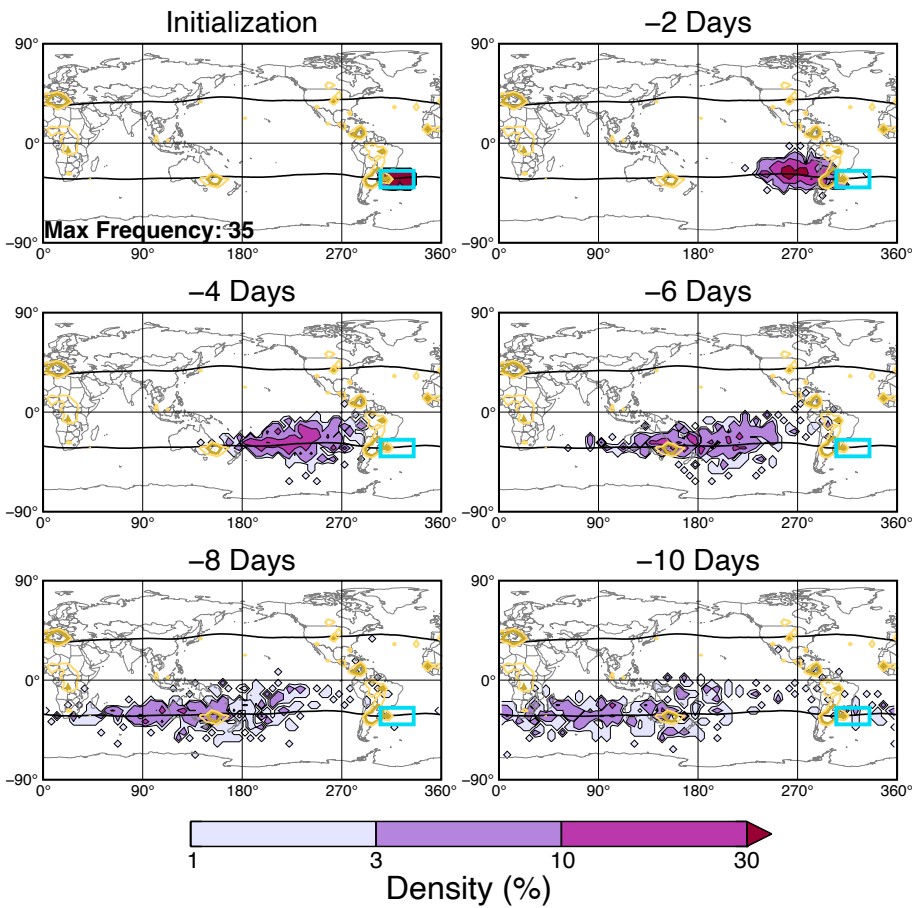

**Figure A2.** As in Fig. 3, but for the SA region in SON.





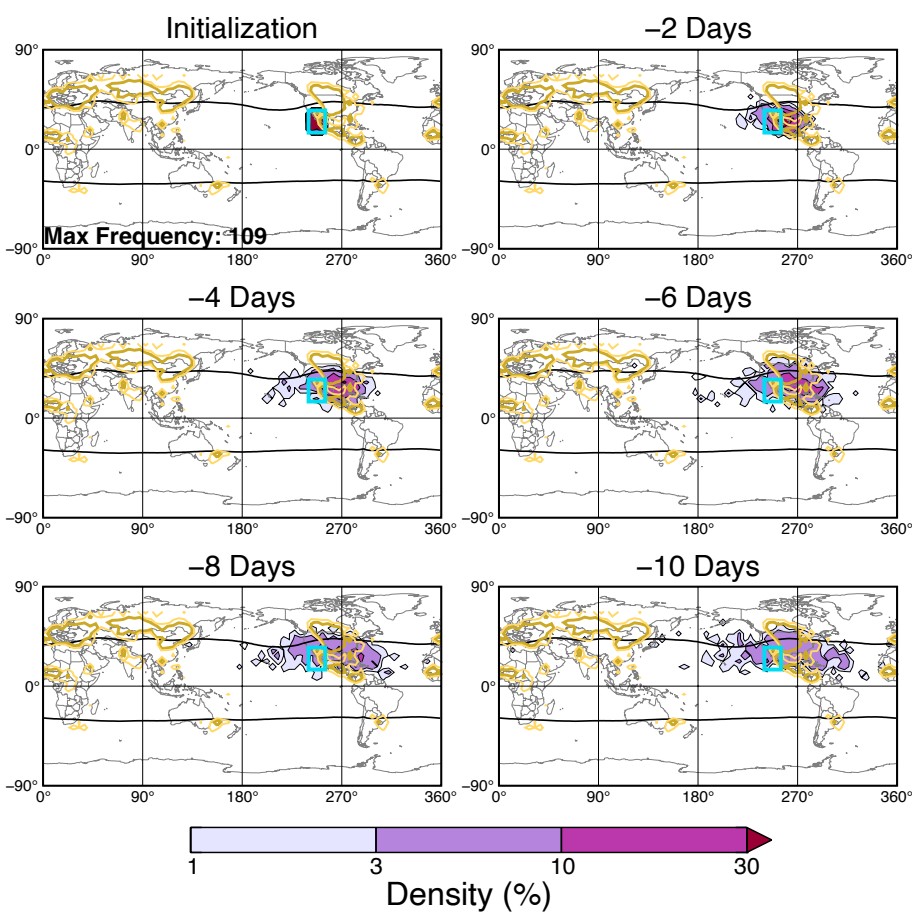

**Figure A3.** As in Fig. 3, but for the GC region in JJA.



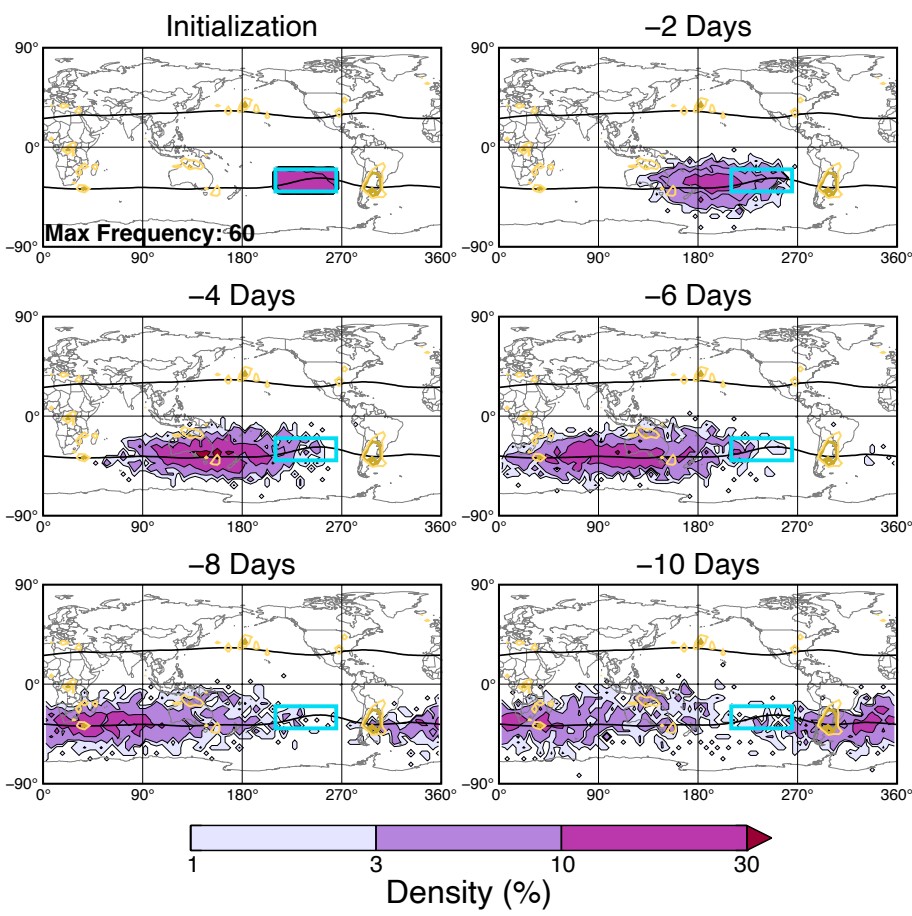

**Figure A4.** As in Fig. 3, but for the SP region in DJF.



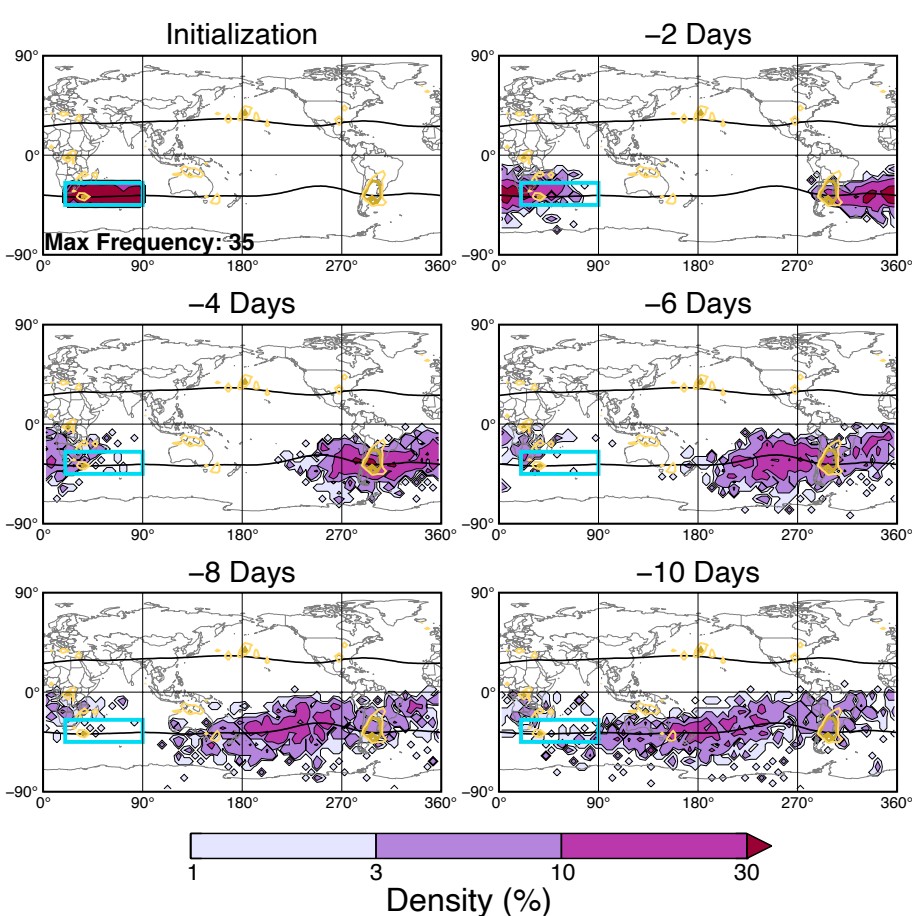

**Figure A5.** As in Fig. 3, but for the SI region in DJF.



*Author contributions.* ET and CH designed the study and ET carried it out. ET wrote the original draft of the manuscript, with review and editing by CH.

*Competing interests.* The authors declare that they have no conflict of interest.

*Acknowledgements.* We thank Nana Liu and Chuntao Liu for providing the processed GPM data used for context in this manuscript. This work was supported by the National Aeronautics and Space Administration (NASA) under Awards 80NSSC18K0746 and 80NSSC19K0347.



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
