# Peer review of "Climatology, sources, and transport characteristics of observed water vapor extrema in the lower stratosphere"

_EGUsphere, 2023_

## Referee Comment (RC2)

**Review of "Climatology, sources, and transport characteristics of observed water vapor extrema in the lower stratosphere," Tinney and Homeyer (2023)**

**General comments:**

The study authored by Tinney and Homeyer presents an interesting and significant analysis of the contributions of water vapor extrema located within the lowermost stratosphere. In the context of extensive prior work examining stratospheric water vapor extrema that relied upon a more conservative criterion, the authors advance an argument that such a strict criterion provided an incomplete understanding of water vapor extrema and that a more nuanced approach to stratospheric water vapor extrema is necessary. The analytical framework and dataset used are appropriate for the scientific questions addressed, and the results support their argument. However, given that the crux of the study rests upon accurately parsing lowermost stratospheric observations from tropospheric, additional description of and support for the methodology presented is needed.

**Specific comments:**

1. As accurately identifying MLS levels that fall within the lowermost stratosphere is critical to the analysis, and the authors have developed an extensive set of filtering criteria, additional details about how these criteria were selected would strengthen the argument. Specifically, explicit details about the rigorous testing and evaluation mentioned in line 103 are needed. Additionally, how sensitive are the results of your analysis to these criteria?

2. Does the absolute threshold of 8 ppmv for identifying water vapor extrema introduce geographic or seasonal biases due to differences in background concentrations that fluctuate?

3. Given the importance of tropopause height to this analysis, are any complications introduced by the use of ERA-5 tropopause height for the GPM data while MERRA-2 tropopause heights are used with the MLS data?

4. Why are annual cycle analyses for the other regions identified in section 3.1 not included?

5. The "Conclusions" section needs a brief discussion of the limitations associated with the assumptions of the study design, and with the various proxies (e.g. 60% of prior 5 days spent in the troposphere as large-scale TST) used.

**Technical corrections:**

Line 22 – "essential to improve understanding" OR "essential for improving our understanding"

Line 24 – this sentence is a bit convoluted and may benefit from being split in two

Line 25 – authors may consider including discussion of water vapor climate feedbacks (e.g. Banerjee et al., 2019; Konopka et al., 2022; Nowack et al., 2023)

Line 27 – theta has not been introduced as potential temperature

Line 30 – is it important to specify how deep into the stratospheric overworld you are defining the 'total LS?'

Line 33 –  additionally

Line 46 – extreme LOCALIZED stratospheric hydration

Line 48 – Clapp et al. 2019 and Liu et al. 2020 also support this

Line 168 –  in contrast

Line 169 – no source according to what?

Line 175 – provide statistics of number of layers and profiles included that would have been excluded using prior criteria

Figure 3 – define tropopause break

Line 212 – what sort of timing of convection and timing of mixing is necessary to create hotspots in the Fig 1 distribution instead of a latitude band smear in the context of zonal flow?

Line 215 – Clapp et al. (2021) discuss significant outflow from the North American Monsoon Anticyclone to the west in this region consistent with the authors' results

Figure 6 – why the 20-profile threshold?

Line 270 – 100 hPa previously defined as stratospheric overworld earlier in the introduction

Line 279 – why is the US the dominant contributor? From the figure it appears equal in magnitude to the stereotypical Southwest US/Mexico NAM region. What contributes to the September and October period?

Line 284 – is this not due to differences in tropopause heights associated with the different monsoons?

Line 290 – why the peak in December/January for AMA and NAMA?

Line 297 – note the percentages as percentages of total MLS observations

Line 299 – "the analysis"

Line 305 – was this not the case for the SC region?

Line 307 – does monsoon circulation not include a meridional component?

Line 309 – reiterate how this study defines relationship to large-scale TST

**References:**

Banerjee, A., Chiodo, G., Previdi, M., Ponater, M., Conley, A.J. and Polvani, L.M., 2019. Stratospheric water vapor: an important climate feedback. *Climate Dynamics*, *53*, pp.1697-1710.

Clapp, C.E., Smith, J.B., Bedka, K.M. and Anderson, J.G., 2019. Identifying source regions and the distribution of cross-tropopause convective outflow over North America during the warm season. *Journal of Geophysical Research: Atmospheres*, *124*(24), pp.13750-13762.

Clapp, C.E., Smith, J.B., Bedka, K.M. and Anderson, J.G., 2021. Identifying Outflow Regions of North American Monsoon Anticyclone-Mediated Meridional Transport of Convectively Influenced Air Masses in the Lower Stratosphere. *Journal of Geophysical Research: Atmospheres*, *126*(10), p.e2021JD034644.

Konopka, P., Tao, M., Ploeger, F., Hurst, D.F., Santee, M.L., Wright, J.S. and Riese, M., 2022. Stratospheric moistening after 2000. *Geophysical Research Letters*, *49*(8), p.e2021GL097609.

Liu, N., Liu, C. and Hayden, L., 2020. Climatology and detection of overshooting convection from 4 years of GPM precipitation radar and passive microwave observations. *Journal of Geophysical Research: Atmospheres*, *125*(7), p.e2019JD032003.

Nowack, P., Ceppi, P., Davis, S.M., Chiodo, G., Ball, W., Diallo, M.A., Hassler, B., Jia, Y., Keeble, J. and Joshi, M., 2023. Response of stratospheric water vapour to warming constrained by satellite observations. *Nature Geoscience*, pp.1-7.

---

## Author Response (AR1)

**Response to Reviewers for EGUSPHERE-2023-985**

We thank the reviewers for their insight and comments that helped us improve the manuscript. Detailed responses to each Reviewer's comments can be found in the following pages with the reviewers' original comments in **black** and our responses in **blue**). All line numbers in the author responses refer to the submitted revised manuscript. For convenience, an additional version of the revised manuscript with bolded track changes is included as an attachment in an additional comment (but note that line numbers may be off slightly due to the sizing of the bold-face text).

**Reviewer #1**

This is a nice analysis and a well-written paper, and will be of interest to ACP readers. I have only a few minor comments and suggestions.

We sincerely thank you for these kind comments!

**Minor comments:**

Choice of 8 ppmv as a threshold for extrema. I know this is similar to what other studies have done (e.g., Schwarz et al) and that the authors results are not sensitive to the exact threshold per their sensitivity analysis, but would there be some value to using a threshold that varies regionally and/or with vertical level (e.g., a threshold of 3 sigma above the mean)? This might lead to better representation of "extreme" values and would also inherently take into account (im)precision of the MLS measurements. Perhaps some additional discussion/motivation or sensitivity analysis on this would be helpful.

This is an intriguing idea that we gave some thought to, but ultimately decided against a sigma-based approach since water vapor extrema diffuse into the local stratosphere and increase the background mean concentration. Over North America, there exists a seasonal cycle in LS water vapor due (at least in part) to this diffusion of convectively influenced air. Figure from Tinney and Homeyer 2021:

[Figure]

A sigma-based approach would, for example, suggest that the convective injection of water in August is less extreme (or common) than the convective injection of water in March, which we believe is in opposition with the ultimate goal of this research: a

climatology of high water vapor observations. As for a more detailed discussion on the implications of the choice of threshold, see our response to Reviewer #2's Specific Comment #2 below.

We have added substantial discussion of the sensitivity of the analysis in Lines 356–364.

Tinney, E. N., & Homeyer, C. R. (2021). A 13-year trajectory-based analysis of convection-driven changes in upper troposphere lower stratosphere composition over the united states. *JGR: Atmospheres*, 126, e2020JD033657. https://doi.org/10.1029/2020JD033657

Line 41: "assent" -> "ascent"

Done.

Lines 47-48: Consider moving parenthetical statement to end of clause, i.e. "… relative frequently over land (…), … "

Done.

Line 58: "studying global" -> "studying the global"

Done.

Line 67: I suggest using LMS here instead of LS to emphasize that LMS is the specific sub-region that is underestimated.

Done.

Line 90-91: I think you are referring specifically to the MLS WV measurements here, not just MLS measurements in general, so you should probably mention that.

Done.

Line 92: I recommend to use "less" here rather than "below" because below is often interpreted as a direction and could be cause confusion.

Great point, done.

Regarding the LMS SC extrema in Fig. 1. I was really confused by this when first mentioned on Line 157 because by your definition the LMS shouldn't exist in the tropics. But you eventually provide an explanation around line 175. Maybe it'd be worth mentioning in section 2.3 that under some rare circumstances the criteria for LMS (340K < theta < 380 K, PV > 6, and 1 km above LRT) can be met in the tropics?

This was a surprising result that initially confused us as well! In Lines 122–123 of the revision, we added "In a few rare circumstances, these criteria can be met within the deep tropics; an example of this is discussed in detail in Section 3.2."

Figure 4: Could some of these trajectories in the NP have come from NA or GC convection 10+ days earlier? There seems like a hint that the air came from all the way around the world at 10 days but perhaps the trajectories are too untrustworthy over that length of time.

This is certainly possible, but the spread of trajectories – due to both the complexities of the patterns (the zonal transport from the west vs the transport that traces back eastward to Central America) as well as trajectory error – prevents us from confidentantly making any such conclusions. Additionally, we have added some discussion to the manuscript of mixing and detectability timescales in Lines 181–184. See our response to Review #2's comment, "Line 212 – what sort of timing of convection and timing of mixing is necessary to create hotspots in the Fig 1 distribution instead of a latitude band smear in the context of zonal flow?" below.

**Reviewer #2**

**General comments:**

The study authored by Tinney and Homeyer presents an interesting and significant analysis of the contributions of water vapor extrema located within the lowermost stratosphere. In the context of extensive prior work examining stratospheric water vapor extrema that relied upon a more conservative criterion, the authors advance an argument that such a strict criterion provided an incomplete understanding of water vapor extrema and that a more nuanced approach to stratospheric water vapor extrema is necessary. The analytical framework and dataset used are appropriate for the scientific questions addressed, and the results support their argument. However, given that the crux of the study rests upon accurately parsing lowermost stratospheric observations from tropospheric, additional description of and support for the methodology presented is needed.

We sincerely thank you for these kind and thorough comments!

**Specific comments:**

1. As accurately identifying MLS levels that fall within the lowermost stratosphere is critical to the analysis, and the authors have developed an extensive set of filtering criteria, additional details about how these criteria were selected would strengthen the argument. Specifically, explicit details about the rigorous testing and evaluation mentioned in line 103 are needed. Additionally, how sensitive are the results of your analysis to these criteria?

   Section 2.3 was reworked to include substantial discussion about this process. These additional details can be found in Lines 104–115 of the revision.

2. Does the absolute threshold of 8 ppmv for identifying water vapor extrema introduce geographic or seasonal biases due to differences in background concentrations that fluctuate?

   The percent difference in the annual number of extrema between a 10 ppmv and an 8 ppmv threshold is shown below. In places where such extrema are relatively frequent (i.e. within one of our eight regions), the number of 8 ppmv extrema is typically approximately double that of the 10 ppmv extrema. The region with the largest percent increase is the CEA region, where 8 ppmv extrema occur two to three times more often than 10 ppmv extrema. Overall, the only substantial geographic bias from the choice of threshold appears to be that a lower threshold preferentially favors the Asian Monsoon region. Given our conclusions and emphasis on how the other regions become more prominent when the LMS is

included within the analysis, this bias does not have a meaningful impact on the qualitative results (other than potentially strengthening our argument).

[Figure]

The normalized annual cycle for the Northern Hemisphere and Southern Hemisphere using 8, 10, and 12 ppmv extrema are shown below, with increasing thickness representing increasing leniency in the thresholds (i.e. 8 ppmv is the thickest, 12 ppmv is the thinnest).

This does indicate a slight bias in the 8 ppmv threshold towards a later peak in extrema frequency (August instead of July).

We have reworded the introductory portion of the Results section (Lines 153–157) and added discussion associated with this in the Conclusions in Lines 356–364 to acknowledge these sensitivities.

[Figure]

3. Given the importance of tropopause height to this analysis, are any complications introduced by the use of ERA-5 tropopause height for the GPM data while MERRA-2 tropopause heights are used with the MLS data?

Since the GPM data are simply providing context for general locations of frequent tropopause-overshooting convection, differences in tropopause heights would have a negligible impact on our analysis. Additionally, any differences in tropopause height would be negligible compared to the coarse vertical resolution of MLS and our conservative requirements, as prior studies have shown they are likely ~100 m (e.g., see Hoffman and Spang 2022).

Hoffmann, L. and Spang, R.: An assessment of tropopause characteristics of the ERA5 and ERA-Interim meteorological reanalyses, Atmos. Chem. Phys., 22, 4019–4046, https://doi.org/10.5194/acp-22-4019-2022, 2022.

4. Why are annual cycle analyses for the other regions identified in section 3.1 not included?

Overall, we feel that the seasonality of the extrema in all regions is well-represented by the seasonal panels in Figure 2. The motivation for this additional section was to take a closer look at extrema that would be substantially impacted by the annual cycles of the monsoon circulations. Additional language to emphasize the motivation here was added in Lines 284–286.

5. The "Conclusions" section needs a brief discussion of the limitations associated with the assumptions of the study design, and with the various proxies (e.g. 60% of prior 5 days spent in the troposphere as large-scale TST) used.

A paragraph focusing on the limitations of such choices (specifically, the large-scale TST proxy and the extrema threshold choice) was added to the Conclusions in Lines 351–364

**Technical corrections:**

Line 22 – "essential to improve understanding" OR "essential for improving our understanding"

Done.

Line 24 – this sentence is a bit convoluted and may benefit from being split in two

Done. The reworded sentence(s) can be found in Lines 25–27 of the revision.

Line 25 – authors may consider including discussion of water vapor climate feedbacks (e.g. Banerjee et al., 2019; Konopka et al., 2022; Nowack et al., 2023)

Great idea! We have added a sentence and these references in Lines 23–25 of the revision.

Line 27 – theta has not been introduced as potential temperature

This has been fixed.

Line 30 – is it important to specify how deep into the stratospheric overworld you are defining the 'total LS?'

We have added language here to specify that the LS defined here is bound by the 450 K isentrope.

Line 33 – alternatively additionally

We feel that the word "alternatively" is appropriate here since we are presenting and emphasizing the contrasting controls on water vapor in the LMS vs the overworld.

Line 46 – extreme LOCALIZED stratospheric hydration

Done.

Line 48 – Clapp et al. 2019 and Liu et al. 2020 also support this

Liu et al. 2020 was already included as a reference in this line, but we have added Clapp et al. 2019 as well.

Line 168 – alternatively in contrast

Done.

Line 169 – no source according to what?

This sentence was slightly reworded and a citation for Liu et al. 2020 was added. The revised sentence can be found at Lines 190–191.

Line 175 – provide statistics of number of layers and profiles included that would have been excluded using prior criteria

We are very appreciative of this comment, as this helped us recognize an oversight in our description of the method. In section 2.3 of the original manuscript, Line 110 incorrectly stated "For analysis, all identified stratospheric MLS layers are collected in 5° latitude-longitude bins". In reality, we only include one stratospheric layer (the one with the highest water vapor concentration) from each MLS profile in the analysis to avoid

double or triple counting a particularly deep extrema feature. For the overworld analysis, we include only the wettest overworld layer in the analysis. Therefore, the counts for the number of profiles included in the total LS, the overworld, and the 100 hPa analysis are nearly identical. We have modified the language in section 2.3 to correct this (Lines 124–125 of the revision).

Figure 3 – define tropopause break

Added "the location of the sharp discontinuity between tropical and extratropical tropopause heights" as the definition.

Line 212 – what sort of timing of convection and timing of mixing is necessary to create hotspots in the Fig 1 distribution instead of a latitude band smear in the context of zonal flow?

Typical mixing timescales in the LMS can be estimated to be 5-7 days (e.g., Homeyer et al. 2011). Using some assumptions, we can estimate the limits of detectability in space and time, but the temporal range of detectability of any individual enhancement will be largely dependent upon the strength of the initial water vapor enhancement (i.e., how much it exceeds the 8 ppmv threshold).

Assuming a timescale of 6 days, we can examine the strength of a water vapor delta as a function of days since it was sourced using: $(\Delta H_2 O)_t = (\Delta H_2 O)_0 * e^{-(t/6)}$

So, for an initial 10 ppmv observation, and assuming a background concentration of 5 ppmv, our $(\Delta H_2 O)_0$ is 5 ppmv, and $(\Delta H_2 O)_t$ must be >=3 ppmv in order for the total concentration to exceed the 8 ppmv threshold used in this study. At 1 day past the initial enhancement, the total water vapor concentration is:

$$(H_2 O)_{1.0} = 5\ ppmv + (\Delta H_2 O)_{1.0} = 5\ ppmv + ((5\ ppmv * e^{-(1.0/6)}) = 9.23\ ppmv$$

and remains detectable in our study. At 3 days past the initial enhancement, the total concentration is:

$$(H_2 O)_{3.0} = 5\ ppmv + (\Delta H_2 O)_{3.0} = 5\ ppmv + (5\ ppmv * e^{-(3.0/6)}) = 8.03\ ppmv$$

and is marginally detectable in our study. So, in this case, if we assume a temporal detectability range of 3 days and typical zonal flow in the stratosphere (U = 25 m/s), a feature would be detectable at:

3 days * 25m/s * 24 hr/day * 3600 s/hr * 0.001 km/m = 6480 km

from its initial location. At 30 degrees N, 6480 km is ~67 degrees longitude.

Again, the range of detectability is largely dependent on the initial water vapor enhancement. An 8.5 ppmv initial concentration would not be detectable within 24 hours, while a 15 ppmv initial concentration could be detectable for over a week.

Obviously, there are a number of assumptions made here and many unconsidered variables (e.g., vertical wind shear) that could impact the resulting range of detectability (largely decreasing the above time estimates), but overall, our features appear physically reasonable given common mixing timescales, differences in water vapor enhancements, and eastward advection in the lower stratosphere. Additionally, this aligns well with our discussion of potential sources of extrema at 4 - 6 days prior.

Brief discussion related to mixing and detectability timescales was added to Lines 181–184 of the revision.

Homeyer, C. R., Bowman, K. P., Pan, L. L., Atlas, E. L., Gao, R.-S., and Campos, T. L. (2011), Dynamical and chemical characteristics of tropospheric intrusions observed during START08, *J. Geophys. Res.*, 116, D06111, doi:10.1029/2010JD015098.

Line 215 – Clapp et al. (2021) discuss significant outflow from the North American Monsoon Anticyclone to the west in this region consistent with the authors' results

Thank you for pointing this out! A reference to this work has been added (Lines 237-238 of the revision).

Figure 6 – why the 20-profile threshold?

This is simply chosen to restrict the analysis to bins where enough extrema observations are present to provide reasonable percentages and visual clarity. Added "To restrict the analysis to bins with sufficient sampling of extrema…" to the caption for clarification.

Line 270 – 100 hPa previously defined as stratospheric overworld earlier in the introduction

This was changed from "upper tropospheric" to "tropopause level", which is more accurate and more in line with discussion within the introduction.

Line 279 – why is the US the dominant contributor? From the figure it appears equal in magnitude to the stereotypical Southwest US/Mexico NAM region. What contributes to the September and October period?

The language of this sentence was confusing. We were intending to say that NAMA is dominant over AMA as the NAMA region encompasses US convection in addition to the

stereotypical monsoon convection – not that US convection is dominant over Sierra Madre/Southwest US convection. The September and October period can additionally be explained by United States convection which is still relatively frequent during this season. This sentence has been reworded to reflect this and can be found in Lines 300–304 of the revision.

Line 284 – is this not due to differences in tropopause heights associated with the different monsoons?

Yes, this is absolutely true and we have added language and a reference to clarify this in Lines 172–173 and Line 308 of the revision. However, we believe the takeaway from this is still the same: it is important to consider the LMS when looking at monsoon impacts on the LS.

Line 290 – why the peak in December/January for AMA and NAMA?

Added "Both the AMA and NAMA regions also experience a relative maxima in the frequency of large-scale TST in December/January, which is likely a result of the substantial decrease in tropopause-overshooting convection in the winter months." in Lines 317–319 of the revision.

Line 297 – note the percentages as percentages of total MLS observations

Done.

Line 299 – "the analysis"

Done.

Line 305 – was this not the case for the SC region?

As the transport pathway for the SC region primarily tracks back to the Asian Monsoon circulation (by ~4 days), where it then remains for the rest of the trajectory period, we do not believe the SC region belongs in the category of "largely zonal transport along the tropopause break via subtropical jet streams".

Line 307 – does monsoon circulation not include a meridional component?

Yes, great catch. We added in "outside of monsoon-related circulations" to this sentence for clarification – which is what we were initially trying to communicate.

Line 309 – reiterate how this study defines relationship to large-scale TST

Added "...by using the percentage of trajectories with recent history within the troposphere as a proxy for large-scale isentropic TST" . See Line 336 of the revision.